# Hybrid Fusion of High-Resolution and Ultra-Widefield OCTA Acquisitions for the Automatic Diagnosis of Diabetic Retinopathy

**DOI:** 10.3390/diagnostics13172770

**Published:** 2023-08-26

**Authors:** Yihao Li, Mostafa El Habib Daho, Pierre-Henri Conze, Rachid Zeghlache, Hugo Le Boité, Sophie Bonnin, Deborah Cosette, Stephanie Magazzeni, Bruno Lay, Alexandre Le Guilcher, Ramin Tadayoni, Béatrice Cochener, Mathieu Lamard, Gwenolé Quellec

**Affiliations:** 1Inserm, UMR 1101 LaTIM, F-29200 Brest, France; 2Univ Bretagne Occidentale, F-29200 Brest, France; 3IMT Atlantique, ITI Department, F-29200 Brest, France; 4Sorbonne University, F-75006 Paris, France; 5Service d’Ophtalmologie, Hôpital Lariboisière, AP-HP, F-75475 Paris, France; 6Carl Zeiss Meditec Inc., Dublin, CA 94568, USA; 7ADCIS, F-14280 Saint-Contest, France; 8Evolucare Technologies, F-78230 Le Pecq, France; 9Service d’Ophtalmologie, CHRU Brest, F-29200 Brest, France

**Keywords:** diabetic retinopathy classification, multimodal information fusion, deep learning, computer-aided diagnosis

## Abstract

Optical coherence tomography angiography (OCTA) can deliver enhanced diagnosis for diabetic retinopathy (DR). This study evaluated a deep learning (DL) algorithm for automatic DR severity assessment using high-resolution and ultra-widefield (UWF) OCTA. Diabetic patients were examined with 6×6 mm${}^{2}$ high-resolution OCTA and 15×15 mm${}^{2}$ UWF-OCTA using PLEX®Elite 9000. A novel DL algorithm was trained for automatic DR severity inference using both OCTA acquisitions. The algorithm employed a unique hybrid fusion framework, integrating structural and flow information from both acquisitions. It was trained on data from 875 eyes of 444 patients. Tested on 53 patients (97 eyes), the algorithm achieved a good area under the receiver operating characteristic curve (AUC) for detecting DR (0.8868), moderate non-proliferative DR (0.8276), severe non-proliferative DR (0.8376), and proliferative/treated DR (0.9070). These results significantly outperformed detection with the 6×6 mm${}^{2}$ (AUC = 0.8462, 0.7793, 0.7889, and 0.8104, respectively) or 15×15 mm${}^{2}$ (AUC = 0.8251, 0.7745, 0.7967, and 0.8786, respectively) acquisitions alone. Thus, combining high-resolution and UWF-OCTA acquisitions holds the potential for improved early and late-stage DR detection, offering a foundation for enhancing DR management and a clear path for future works involving expanded datasets and integrating additional imaging modalities.

## 1. Introduction

### 1.1. Context

Diabetic retinopathy (DR), the most frequent complication of diabetes, is a primary cause of blindness in working-age people [1,2]. There are approximately 285 million people who are affected by DR worldwide [3]. Projections indicate that this number will swell to approximately 454 million by the year 2030 [4].

The field of ophthalmology has seen remarkable advancements in retinal imaging technology, which now plays a crucial role in the clinical diagnosis of DR. As a major advancement, optical coherence tomography (OCT) has been a game-changer since its introduction in 1991. OCT has transformed not only the evaluation of the retina but the entire field of ophthalmology [5]. Based on OCT’s foundations, optical coherence tomography angiography (OCTA) offers a non-invasive method for producing detailed and depth-resolved images of the chorioretinal microvasculature. The technique works by analyzing differences between two scans taken at the same location. Moving structures, such as red blood cells, generate a decorrelation signal. Thus, by detecting these signals, OCTA can highlight the retinal vascular networks, offering a rich picture of the retina’s health [6].

Recently, swept-source technology has been used in OCTA, leading to the development of swept-source OCTA (SS-OCTA). This new approach, lauded for its non-invasive, safe, and repeatable imaging of retinal blood flow, has been the subject of numerous studies exploring its potential in diagnosing, screening, and monitoring DR [7,8,9,10]. The technological leap from SD-OCTA to SS-OCTA allowed imaging larger fields of view: most of initial studies used SS-OCTA equipment that could capture a 12×12 mm${}^{2}$ area in a single scan (as opposed to the previous typical areas of 3×3 mm${}^{2}$ or 6×6 mm${}^{2}$) [9,11,12]. This imaging area can be further expanded by stitching together multiple scans or adding dioptric lenses [7,10,13,14], although these techniques may require longer acquisition times and are likely to introduce more artifacts [15]. Machines have recently been developed that can obtain 15×15 mm${}^{2}$ or wider retinal blood flow images by a single scan to effectively solve these problems and provide a fast, reliable solution for DR diagnosis and screening [16,17]. The introduction of ultra-widefield SS-OCTA (UWF-SS-OCTA) offered a broader view for assessing DR lesions [18].

The early detection and timely treatment of DR play a critical role in preventing blindness. However, as the global diabetic population expands, a larger number of qualified ophthalmologists is required to meet the growing demand for DR detection [3]. In response to this challenge, developing automated methods for DR detection has become a priority. The use of deep learning algorithms has recently taken off as a powerful tool for automating or assisting in the diagnosis of DR [19]. Particularly, convolutional neural networks (CNNs) have been demonstrated to be capable of detecting DR in OCTA images [20,21]. Additionally, CNN fusion networks of OCTA structural and flow information can improve the accuracy of DR diagnosis [22]. However, to date, no research has been conducted on the fusion of images obtained from multiple OCTA acquisitions, and herein, we investigated the accuracy of a deep learning algorithm for the automatic assessment of DR severity using high-resolution and ultra-widefield OCTA acquisitions.

### 1.2. OCTA Acquisitions

In this study, we used high-resolution 6×6 mm${}^{2}$ SS-OCTA and 15×15 mm${}^{2}$ UWF-SS-OCTA images obtained from a PLEX® Elite 9000 (Carl Zeiss Meditec Inc., Dublin, CA, USA) for the diagnosis of DR. Each OCTA image encompassed both structural (Structure) and flow (Flow) information. It has been demonstrated in several studies that combining structural and flow information can improve DR diagnosis accuracy [23,24].

The 6×6 mm${}^{2}$ high-resolution SS-OCTA provides superior visualization of the capillary network and the central avascular zone [25]. Consequently, it enables the calculation of metrics such as vascular density (the ratio of vessel area with respect to the total area) [26,27,28], fractal dimensions [29], and intercapillary spaces [30]. However, its limitation lies in its focus on the macular region, potentially neglecting global retinal damages.

On the other hand, the 15×15 mm${}^{2}$ UWF-SS-OCTA provides a more extensive view of the retina, allowing the detection of relevant abnormalities, such as the presence of an intraretinal microvascular abnormality (IRMA) or a preretinal vascular anomaly (neovessel) [11,14,31]. Furthermore, the absence of capillary networks on important surfaces in the 15×15 mm${}^{2}$ image can be easily observed [32], which are considered an important biomarker of proliferating diabetic retinopathy [33,34].

Overall, 6×6 mm${}^{2}$ SS-OCTA allows an accurate calculation of certain vascular metrics and an analysis of the central avascular zone. Still, it only explores a small part of the retina, while 15×15 mm${}^{2}$ SS-OCTA allows a broader investigation of vascular anomalies and areas of non-perfusion in the retina. The two specifications complement each other quite well in clinical practice.

### 1.3. Highlights

This paper presents an innovative approach to improve the accuracy of DR diagnosis by leveraging the complementary information provided by 6×6 mm${}^{2}$ and 15×15 mm${}^{2}$ SS-OCTA images. We meticulously investigated the use of the information from each acquisition and tested the performance of the fusion on structural and flow information for OCTA. Our proposed hybrid fusion network utilizes the structural and flow information of each acquisition, as well as fusing images from both acquisitions to significantly enhance DR diagnostic performance. As the first paper exploring the fusion of different OCTA acquisitions using deep learning methods, this work paves the way for future diagnosis applications from OCTA images.

## 2. Materials and Methods

### 2.1. Hybrid Fusion Workflow

This study aimed to find the best hybrid fusion network structure for the fusion of 6×6 mm${}^{2}$ SS-OCTA data with 15×15 mm${}^{2}$ SS-OCTA data. To achieve this, we organized the workflow into the following four stages:(1)Data processing. The first step involved exploring a variety of approaches to process the raw data from different acquisitions and adapt it to the input specifications of the CNN network.(2)Backbones. Subsequently, we investigated the most effective backbone for the Structure and Flow separately for both acquisitions of OCTA data.(3)Fusion of Structure and Flow. After selecting the most effective backbone from three deep learning architectures for each modality, we evaluated four different fusion strategies—input fusion, feature fusion, decision fusion (leveraging averaging strategies), and hierarchical fusion using Structure and Flow.(4)Fusion of 6×6 mm${}^{2}$ and 15×15 mm${}^{2}$ acquisitions. Based on the best optimal fusion structure for each acquisition, we assessed two strategies, namely feature fusion and decision fusion, on information derived from both 6×6 mm${}^{2}$ and 15×15 mm${}^{2}$ SS-OCTA acquisitions:For the feature fusion strategy, we utilized the model parameters obtained in the previous fusion step and conducted two types of fine-tuning—(a) fine-tuning the entire network (network fine-tuning), and (b) freezing all convolutional layers and fine-tuning the classification layer (layer fine-tuning).For the decision fusion strategy, we implemented and tested both averaging (Avg) and maximization (Max) strategies.

This comprehensive process led us to a hybrid fusion network structure that facilitates the fusion of single-acquisition multimodal information with multiple-acquisition information. This hybrid fusion structure maximized the diagnosis performance of DR by integrating the complementary information from both 6×6 mm${}^{2}$ and 15×15 mm${}^{2}$ SS-OCTA acquisitions. This method fully leveraged the structural and flow information derived from each acquisition, thus optimizing our diagnosis process. Figure 1 illustrates the workflow of this study.

### 2.2. Data Processing

#### 2.2.1. EviRed Dataset

Data regarding DR provided by the Évaluation Intelligente de la Rétinopathie diabétique (EviRed) project (https://evired.org/—accessed on 24 August 2023) were used in this study. Patient data were collected between 2020 and 2022 from 14 hospitals and recruitment centers in France using a PLEX® Elite 9000. The examination was conducted with patients’ informed consent. The Declaration of Helsinki was followed during all procedures. The study protocol was approved by the French South-West and Overseas Ethics Committee 4 on 28 August 2020 (Clinical Trial NCT04624737). The PLEX® Elite 9000 has a scanning frequency of 200 kHz and is capable of acquiring both 15×15 mm${}^{2}$ and 6×6 mm${}^{2}$ SS-OCTA images with a wavelength of 1060 nanometers. In the early phase of the EviRed project, OCTA data were gathered for 875 eyes from a total of 444 patients without a quality filter. This substantial dataset was used to train and test our deep learning models.

Following the EviRed study protocol, each patient’s ocular data often contained two specifications of acquisitions: 6×6 mm${}^{2}$ high-resolution SS-OCTA and 15×15 mm${}^{2}$ UWF-SS-OCTA. Figure 2 shows en-face images and their corresponding B-scan images (pre-processed) of the Structure and Flow from the same patient acquired for different specifications.

The EviRed raw data size was 500×1536×500×2 voxels for the 6×6 mm${}^{2}$ SS-OCTA and 834×3072×834×2 voxels for the 15×15 mm${}^{2}$ SS-OCTA. The last channel presented the information of Structure and Flow, respectively. To reduce the volume under consideration, we intercepted the OCTA image located between the internal limiting membrane (ILM) and retinal pigment epithelium (RPE) layers along the depth (y axis) and flattened the ILM layer. The EviRed raw data were resized to dimensions of 500×224×500×2 voxels for the 6×6 mm${}^{2}$ SS-OCTA and 834×224×834×2 voxels for the 15×15 mm${}^{2}$ SS-OCTA. Figure 2a,c illustrate the orientation of each dimension. For the 6×6 mm${}^{2}$ SS-OCTA, the en-face images had a size of 500×500 pixels, and the B-scan images were 500×224 pixels. The 15×15 mm${}^{2}$ SS-OCTA had en-face images and B-scan images of sizes 834×834 pixels and 834×224 pixels, respectively.

#### 2.2.2. OCTA Cropping

Due to graphics processing unit (GPU) hardware limitations (NVIDIA Tesla V100S with 32 GB memory), our 3D deep learning backbones could only accommodate inputs up to 224×224×224×2 voxels. The patch extraction method is commonly used to address hardware limitations in 3D medical images [35,36]. Nevertheless, it is difficult to ensure that each patch contains pathology information. Based on the idea of test time augmentation [37], the model synthesized and analyzed multiple predictions in order to avoid making inaccurate predictions. As a result, a global prediction of multiple patches was an effective method under the limitations of our hardware. In this context, we proposed a strategy, named *N* times Random Crop method, for processing images as shown in Figure 3. We compared our proposed method with other commonly used methods of data processing. For this comparison, we used the input fusion of ResNet [38] with the 15×15 mm${}^{2}$ OCTA in order to verify its effectiveness. The following methods were tested for prediction:(1)*N* times Random Crop (proposed). During the training of the deep learning network, Random Crop processing was employed, while in the prediction process, we utilized multiple volumes extracted from the OCTA image (*N* times Random Crop) simultaneously to make predictions. Considering that the patch size was 224×224×224×2 voxels, it would take at least 9 batches ($\left\lceil \frac{500}{224}\right\rceil \times \left\lceil \frac{224}{224}\right\rceil \times \left\lceil \frac{500}{224}\right\rceil \times \left\lceil \frac{2}{2}\right\rceil$) to traverse the 500×224×500×2 voxel 6×6 mm${}^{2}$ SS-OCTA images, while 16 batches ($\left\lceil \frac{834}{224}\right\rceil \times \left\lceil \frac{224}{224}\right\rceil \times \left\lceil \frac{834}{224}\right\rceil \times \left\lceil \frac{2}{2}\right\rceil$) would be required to traverse the 834×224×834×2 voxel 15×15 mm${}^{2}$ SS-OCTA images. By comparing the performance of the ResNet input fusion model on the validation set with different *N* times Random Crop methods, we determined the *N* values for the two SS-OCTA acquisitions. The final prediction for an OCTA image was based on the severest prediction among these *N* predictions.(2)Resize. This method compressed the original volume of 834×224×834×2 voxels into 224×224×224×2 voxels for both training and prediction.(3)Center Crop. This approach selected a random patch of 224×224×224×2 voxels from the original 834×834×834×2 voxel OCTA for training. For prediction, a central patch was selected.(4)Subvolume Crop. This technique traversed the OCTA using a window, predicting all subvolumes of 224×224×224×2 voxels and determining the maximum value.

It is worth noting that for single-acquisition fusion, we ensured the registration of data across different modalities. However, when fusing data from different acquisitions, Random Crop generated data from varying regions. Having processed the data, our next step was to use these images to extract meaningful features and combine them for our classification task.

### 2.3. Multimodal Information Fusion

In this section, we describe three fusion network structures commonly used in multimodal research: input fusion, feature fusion, and decision fusion [39]. Furthermore, we introduce hierarchical fusion, which is our extension of traditional feature fusion.

#### 2.3.1. Input Fusion

Input fusion refers to the combination of multiple modalities into a single data tensor, which is then fed into a deep neural network, as illustrated in Figure 4a. It is common to treat different modalities as different input channels when combining modalities with similar structures. This method is widely used in multiple-sequence classification [40,41,42,43] and segmentation [44,45,46,47,48] applications due to its simple implementation. Despite the data fusion at the input level and the single-branch feature extraction structure, complementary information from different modalities is not fully exploited. In addition, input fusion often requires the registration of different input modalities [22].

#### 2.3.2. Feature Fusion

Feature fusion is achieved using different deep learning backbones to extract features from different modalities separately, followed by a fusion process before a final decision is made by the fully connected (FC) layer, as shown in Figure 4b. The feature fusion process uses different branches to extract features and fuse information at the high-dimensional feature level, which is suitable for unregistered data or data with different dimensions [49,50,51,52]. Feature fusion consists merely of concatenating high-dimensional features, which causes relevant information to be lost inadvertently, negatively impacting classification accuracy [22].

#### 2.3.3. Decision Fusion

Decision fusion involves extracting features and making decisions through separate deep learning backbones, and the results are combined into one final decision, as shown in Figure 4c. Many fusion strategies have been proposed for decision fusion [53]. Most of them are based on averaging and majority voting [54,55]. Due to the absence of feature fusion, it is difficult to exploit the complementary information between different modalities.

#### 2.3.4. Hierarchical Fusion

Our previous study proposed hierarchical fusion through the extension of feature fusion [22]. Similar to feature fusion, hierarchical fusion extracts individual features from multiple deep learning branches and then fuses them at higher levels of the network. On the other hand, unlike feature fusion, additional branches are added in order to fuse features at different scales. Finally, the decision layer is applied to the fusion results in order to reach a final prediction. In hierarchical fusion, complementary information among modalities is exploited at different scales, leading to better multimodal fusion. In [22], hierarchical fusion proved to be superior to input fusion and feature fusion in fusing 2D line-scanning ophthalmoscope (LSO), 3D structural OCT, and 3D OCTA images for DR detection. Figure 5 illustrates the hierarchical fusion of the structural information and flow information of the 6×6 mm${}^{2}$ images tested in this study.

### 2.4. Classification Tasks

DR severity was assessed by a retina specialist using fundus photographs, according to the International Clinical Diabetic Retinopathy Disease Severity Scale (ICDR): the absence of diabetic retinopathy, mild nonproliferative diabetic retinopathy (NPDR), moderate NPDR, severe NPDR, proliferative diabetic retinopathy (PDR), and panretinal photocoagulation (PRP). In addition to the six-category multiclass classification, we also performed four binary classification tasks: task0 (detecting mild NPDR or more), task1 (detecting moderate NPDR or more), task2 (detecting severe NPDR or more), and task3 (detecting PDR or PRP). To assess the performance of the four binary classifications, we used the area under the ROC curve (AUC): AUC0 (≥mild NPDR), AUC1 (≥moderate NPDR), AUC2 (≥severe NPDR) and AUC3 (≥PDR). As a standard evaluation metric for the multicategory classification task, Cohen’s kappa was also used to evaluate the EviRed dataset’s six-category results. Based on the confusion matrix, the Kappa coefficient was calculated with a value between −1 (worse than chance agreement) and 1 (perfect agreement).

The calculation formula for the Kappa coefficient based on the confusion matrix is as follows:(1)$$\kappa =\frac{p_{0}-p_{e}}{1-p_{e}},$$
where $p_{0}$ is the accuracy, and $p_{e}$ the hypothetical probability of chance agreement.

### 2.5. Dataset Splitting

During the data acquisition process, there were instances when the collection of OCTA data for both eyes of each patient could not be guaranteed due to factors such as operator errors or the patient’s physical condition. Similarly, not all patients were able to provide both 6×6 mm${}^{2}$ and 15×15 mm${}^{2}$ SS-OCTA data. Despite these constraints, and in order to make full use of the dataset to train different model frameworks for different acquisitions and to test the performance of the fusion model, we split the data as follows: Initially, we selected 53 patients out of the 444 in the EviRed dataset who had both 6×6 mm${}^{2}$ and 15×15 mm${}^{2}$ SS-OCTA data in each eye to form a test set. The remaining patients were shared out randomly between a training set and a validation set. Depending on the fusion task, subsets of the training and validation sets were used in each experiment—all 6×6 mm${}^{2}$ acquisitions or all 15×15 mm${}^{2}$ acquisitions for single-acquisition tasks, all matched pairs of 6×6 mm${}^{2}$ and 15×15 mm${}^{2}$ acquisitions for multiple-acquisition tasks. All fusion tests were trained and validated using five-fold cross-validation (four-fold training and one-fold validation), and performance scores were derived from the same test sets. In the training, validation, and test sets, the distribution of data was identical to the original distribution. The patient and eye data statistics for different fusion datasets are shown in Table 1, and the severity distribution is displayed in Table 2.

### 2.6. Implementation Details

The experiments were carried out with 3D versions of ResNet50 [38], DenseNet121 [56], and EfficientNetB0 [57] trained from scratch. To enhance the robustness of these models, data augmentation techniques such as random Gamma transformations, Gaussian noise injection, and image flipping were employed. For model training, we utilized the Adam optimizer for gradient descent with an initial learning rate of $1\times 10^{-4}$. ExponentialLR with a gamma equal to 0.99 was the learning rate decay strategy. The number of training epochs was set to 500, and the batch size was set to 2. The network training and testing were carried out using four NVIDIA Tesla V100S units with 32 GB memory. For training large models such as the hierarchical fusion used in this experiment, model parallelism was used. The validation set was used to select the best backbones and the best checkpoint of each backbone. It was also used to select the best data cropping and information fusion strategies. However, for simplicity, performance is illustrated solely on the test set hereafter.

## 3. Results

### 3.1. Data Cropping

Figure 6 shows the test results for different *N* times Random Crop methods. The performance of the fusion model on different metrics improved with an increase in *N*. For the 6×6 mm${}^{2}$ SS-OCTA, the performance at *N* = 10 and *N* = 12 was comparable. For the 15×15 mm${}^{2}$ SS-OCTA, the performance at *N* = 20 and *N* = 25 was essentially unchanged. As a result, we chose *N* = 10 for 6×6 mm${}^{2}$ SS-OCTA and 20 for 15×15 mm${}^{2}$ SS-OCTA as reasonable tradeoffs between computation times and classification scores.

Table 3 compares cropping methods: it shows that the Resize and Center Crop methods performed poorly due to a significant loss of information. As a result of the data compression of Resize, many pathological details were rendered invisible, while Center Crop focused only on the information obtained from the center patch. Although Subvolume Crop performed relatively well, manually extracting subvolumes may have omitted key pathological features, affecting the model’s judgment. In the validation and test sets, our proposed data cropping method, namely Random Crop, outperformed the others in each classification task, demonstrating its effectiveness in handling the large original volume of OCTA images.

### 3.2. Backbones

A total of four modalities of information from Structure and Flow with different acquisitions were tested, along with three deep learning backbones: ResNet, DenseNet, and EfficientNet. Table 4 presents the results of the backbone tests. In the validation and test sets, ResNet demonstrated superior performance across all classification tasks for both the structure modality from the 6×6 mm${}^{2}$ SS-OCTA images and the flow modality from the 15×15 mm${}^{2}$ SS-OCTA images. The performance of the other backbones varied across the remaining modalities, and it was difficult to determine which backbone was the most effective. EfficientNet was effective for the multiclass classification as well as early pathology detection in the Flow from 6×6 mm${}^{2}$ SS-OCTA images, while ResNet excelled in the more severe pathology detection tasks. Interestingly, DenseNet surpassed ResNet on task 0 when using Structure from 15×15 mm${}^{2}$ SS-OCTA images. Based on these results, we selected the best-performing backbones (in bold in Table 4) for different tasks as baselines for the subsequent fusion schemes of Structure and Flow.

### 3.3. Fusion of Structure and Flow

We combined Structure and Flow from different acquisitions using the top-performing backbones from the previous section. We tested input fusion, feature fusion, and hierarchical fusion. Table 5 and Table 6 show the fusion results for 6×6 mm${}^{2}$ and 15×15 mm${}^{2}$ SS-OCTA acquisitions, respectively.

In the validation and test sets, the hierarchical fusion outperformed other methods for 6×6 mm${}^{2}$ OCTA. Based on two ResNet branches, the hierarchical fusion method achieved a Kappa value of 0.4752 for the six-category multiclass classification, a significant improvement over the unimodal baseline. Furthermore, hierarchical fusion improved diagnostic performance for both task 0 and task 1. In contrast, hierarchical fusion did not perform as well as unimodal fusion in tasks 2 and 3. There was a significant difference in performance between Structure and Flow in tasks 2 and 3. As a result, fusion was not effective, since Flow did not provide additional complementary information to Structure. Also, the hierarchical fusion of ResNet and EfficientNet was not effective, likely due to the structural differences between these backbones.

Similarly, for the 15×15 mm${}^{2}$ SS-OCTA acquisitions, hierarchical fusion was the most effective in the validation and test sets. The hierarchical fusion of two ResNet branches significantly improved performance for six-category multiclass classification and tasks 1, 2, and 3 compared to the unimodal baseline results. Specifically, hierarchical fusion achieved an AUC of 0.8786 for task 3. Due to the similar performance of Structure and Flow, the hierarchical fusion was able to take advantage of the complementary information provided by the different modalities and performed well.

From the above results, the 6×6 mm${}^{2}$ SS-OCTA was very effective for diagnosing early diabetic retinal lesions, while the 15×15 mm${}^{2}$ SS-OCTA was more effective in diagnosing more advanced pathology, which is consistent with our clinical prior knowledge. As shown in [22], hierarchical fusion proves effective since the Structure and Flow-based hierarchical fusion has the capability of utilizing complementary information to enhance the strengths of each acquisition individually, thereby facilitating the subsequent fusion of different acquisitions.

### 3.4. Fusion of 6×6 mm${}^{2}$ SS-OCTA and 15×15 mm${}^{2}$ SS-OCTA

To maximize the complementary strengths of the 6×6 mm${}^{2}$ SS-OCTA and 15×15 mm${}^{2}$ SS-OCTA acquisitions for different tasks, we further tested feature fusion and decision fusion on the hierarchical fusion architectures. The unimodal results of the 6×6 mm${}^{2}$ SS-OCTA and 15×15 mm${}^{2}$ SS-OCTA images were used as baselines. The results of this fusion are shown in Table 7.

Decision fusion with average aggregation was the best strategy for the validation and test sets. The process of feature fusion showed improvements over single acquisitions on certain tasks but did not achieve its goal for tasks 2 and 3. This discrepancy could have been due to the differing volumes of the acquisitions. Without utilizing registered image information for random cropping, the pathological features between acquisition branches could vary significantly, which could have potentially impaired the judgment of the fusion model. Conversely, decision fusion had the capacity to address this issue effectively.

The decision fusion process operated only on the final output probability after each branch had independently made its assessments. This allowed for the integration of information without being affected by image registration at the same time. As shown in Table 7, the proposed decision fusion method, which was based on averaging, performed well.

The inference time of the different fusion methods was also compared. Inference took longer for 15×15 mm${}^{2}$ SS-OCTA (*N* = 20), since the *N* times Random Crop was twice as large as for 6×6 mm${}^{2}$ SS-OCTA (*N* = 10). Due to the complexity of its structure, hierarchical fusion required more time for inference. Despite this, because of the parallel nature of the model, our hybrid fusion method did not take longer than hierarchical fusion. The resulting four second inference time per eye is acceptable and can provide reliable results for ophthalmologists within a short period of time.

## 4. Discussion and Conclusions

This study investigated a deep learning algorithm to classify diabetic retinopathy severity using 6×6 mm${}^{2}$ high-resolution SS-OCTA and 15×15 mm${}^{2}$ UWF-SS-OCTA acquisitions. It relied on a hybrid fusion architecture that utilized complementary structure and flow information from both acquisitions. In detail, this architecture combined hierarchical fusion to jointly analyze Flow and Structure from the same acquisition and decision fusion to merge predictions from both acquisitions. This algorithm was evaluated on preliminary data from the EviRed project.

Our experiments showed that the 6×6 mm${}^{2}$ SS-OCTA acquisitions were highly effective for the detection of early-stage pathology, while 15×15 mm${}^{2}$ SS-OCTA acquisitions performed better in terms of advanced pathology detection (see Table 7). This was consistent with the perceived usefulness of these acquisitions by ophthalmologists: in the early stages, anomalies are generally small and are therefore better seen in high-resolution SS-OCTA images, while in the advanced stages, anomalies are larger, and an ultra-widefield image becomes more beneficial than a high-resolution image. The suggested hybrid fusion system demonstrated significant improvements over single acquisitions (see Table 7). The hybrid fusion approach integrated the strengths of both acquisitions: it delivered excellent performance in both early and late pathological diagnosis while significantly improving the accuracy of the six-category multiclass classification. Therefore, this study clearly validated the relevance of jointly analyzing multiple acquisitions. To a lesser extent, this study also validated the relevance of analyzing multiple modalities: combining Flow and Structure always outperformed analyzing a single modality, although the performance gain was limited (see Table 5 and Table 6).

In recent times, transformer-based models [58] have shown good performance on classification tasks, such as the Vision Transformer (ViT) [59], which we also tested. The performance of the structure and flow modalities of 6×6 mm${}^{2}$ SS-OCTA images was tested using 3D ViT models (patch size = (32, 32, 32)) from the Monai library (https://monai.io/—accessed on 24 August 2023). Table 8 illustrates the test results for ViT.

In all tasks, ViT performed very poorly. A large dataset and pre-trained models contribute significantly to the excellent performance of ViT [59]. In addition to the limited number of patients in our dataset, there was no publicly available pre-training model for 3D ViT, which was likely the major reason for its poor performance. Nevertheless, extensive testing is still required for the hyperparameter configuration of 3D transformer models.

It should be noted, however, that some transformer-based models are increasingly used to perform multimodal tasks in the medical field [60,61,62]. It has been observed that these models often combine a CNN structure with a transformer structure, resulting in excellent classification performance with limited medical datasets; this is one of the directions that we plan to pursue in the future.

One limitation of this study was that the current dataset is insufficiently large, resulting in suboptimal performance on the six-category multiclass classification task. Furthermore, too small a dataset may adversely affect the robustness of a model. The EviRed project is expected to collect clinical data from thousands of patients, and so more datasets will be tested in the near future. Further studies will be conducted to test the stability of the model and fine-tune the model to improve its performance on the six-category multiclass classification task.

The current EviRed dataset also contains ultra-widefield color fundus photography (UWF-CFP) data alongside OCTA data from different acquisitions, which may aid in further improving the accuracy of DR diagnosis. In [15], the use of UWF-OCTA in conjunction with UWF-CFP was recommended for the screening and follow-up of DR. Conversely, UWF-OCTA alone had some limitations. It is difficult and sometimes ambiguous to identify microaneurysm and intraretinal hemorrhage from OCTA en-face images. To facilitate diagnosis, it is often necessary to search for corresponding lesions on B-scan images, a time-consuming process. The use of UWF-CFP images would make this task much easier. Our further investigation will be directed by the joint analysis of OCTA and UWF-CFP images. The EviRed projects are also aiming to collect longitudinal data. Once enough longitudinal data are available, the proposed framework will be applied to DR prognosis tasks, for the purpose of improving DR management.

## Figures and Tables

**Figure 1 diagnostics-13-02770-f001:**
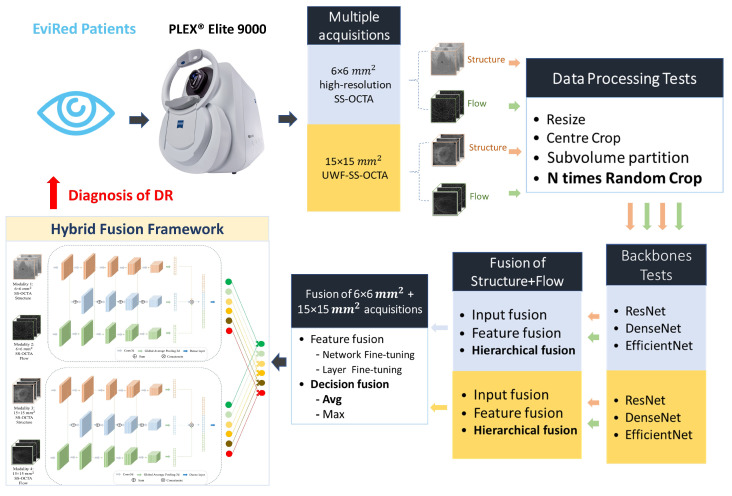
Proposed workflow.

**Figure 2 diagnostics-13-02770-f002:**
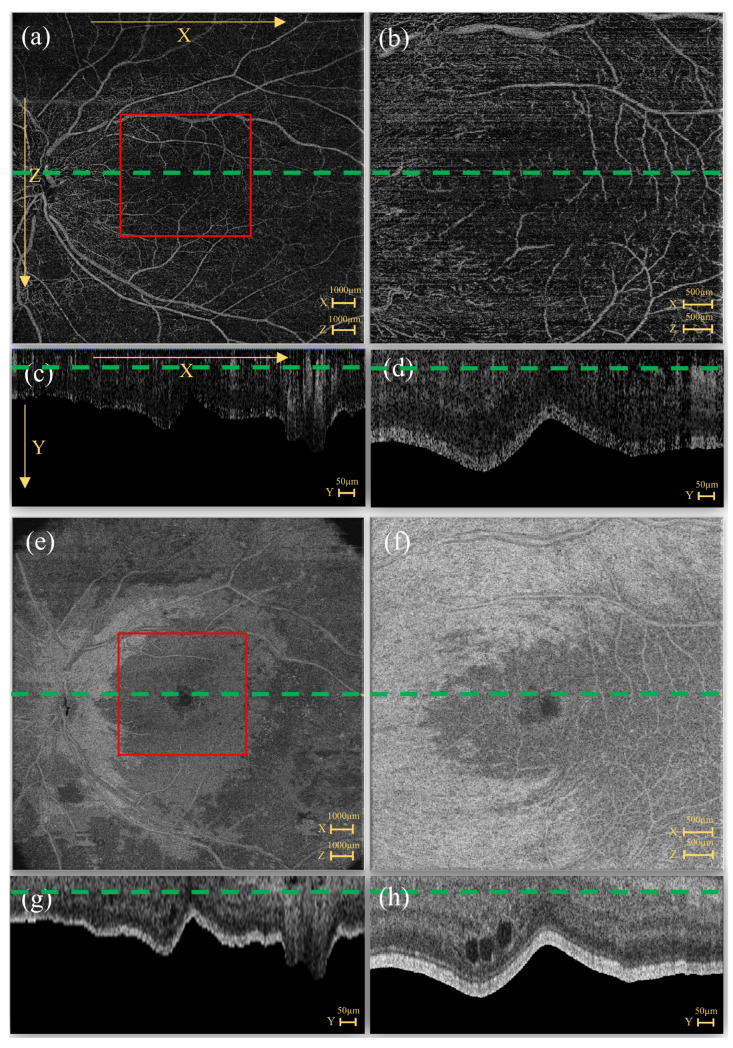
Structure and Flow en-face slices (**a**,**b**,**e**,**f**) and pre-processed B-scan images (flattened retina) (**c**,**d**,**g**,**h**) from 6×6 mm${}^{2}$ SS-OCTA and 15×15 mm${}^{2}$ SS-OCTA. (**a**,**c**) Flow of 15×15 mm${}^{2}$ SS-OCTA. (**b**,**d**) Flow of 6×6 mm${}^{2}$ SS-OCTA. (**e**,**g**) Structure of 15×15 mm${}^{2}$ SS-OCTA. (**f**,**h**) Structure of 6×6 mm${}^{2}$ SS-OCTA. The area of the 6×6 mm${}^{2}$ SS-OCTA is in the center of the 15×15 mm${}^{2}$ SS-OCTA image (red bounding box). The green line in the en-face slice shows the source of the B-scan, and the green line in the B-scan image shows the intercept direction of the en-face slice.

**Figure 3 diagnostics-13-02770-f003:**
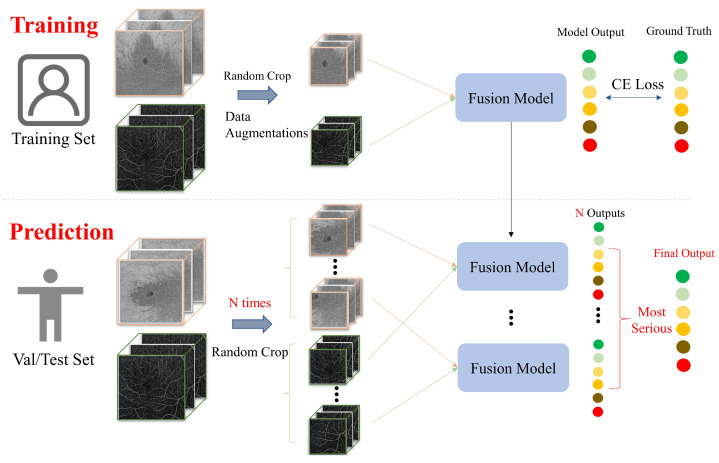
Our proposed data processing approach, where *N* is 10 for 6×6 mm${}^{2}$ SS-OCTA and 20 for 15×15 mm${}^{2}$ SS-OCTA. Predictions were based on the same fusion model as for training. Colored discs indicate the DR severity categories.

**Figure 4 diagnostics-13-02770-f004:**
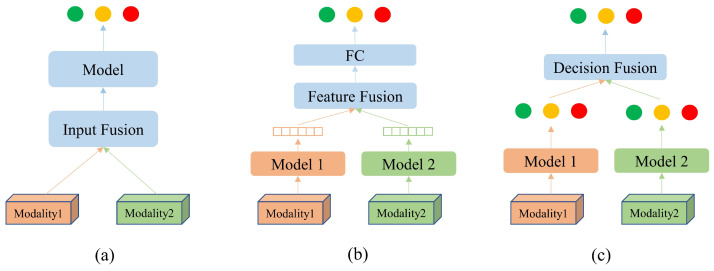
An illustration of the three types of multimodal fusion networks: (**a**) input fusion, (**b**) feature fusion, (**c**) decision fusion.

**Figure 5 diagnostics-13-02770-f005:**
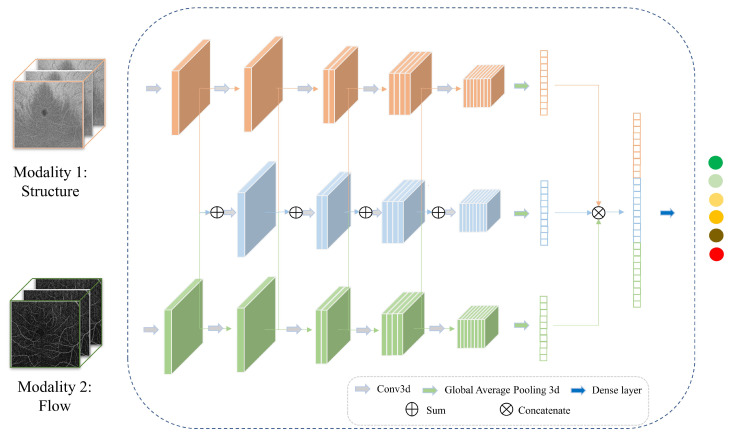
An illustration of hierarchical fusion network for 6×6 mm${}^{2}$ SS-OCTA Structure and Flow.

**Figure 6 diagnostics-13-02770-f006:**
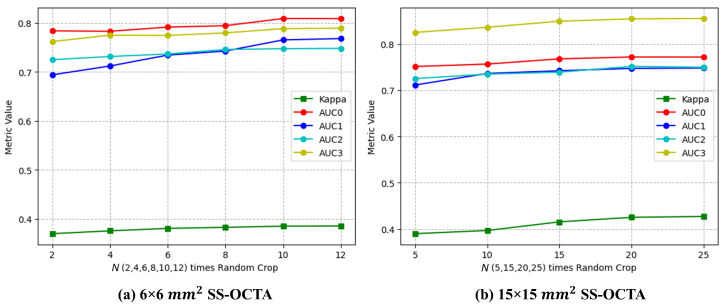
The results of the different *N* times Random Crop methods on the validation set for the input fusion of ResNet with the two SS-OCTA acquisitions.

**Table 1 diagnostics-13-02770-t001:** Statistics on the number of patients and eyes in the dataset. For the fusion of Structure and Flow for 6×6 mm${}^{2}$ SS-OCTA, the fusion of Structure and Flow for 15×15 mm${}^{2}$ SS-OCTA, and the fusion of 6×6 mm${}^{2}$ and 15×15 mm${}^{2}$ SS-OCTA, the test sets were identical and fixed. Dataset 6 × 6, dataset 15 × 15, and dataset 6 × 6 + 15 × 15 represent the corresponding training and validation sets.

Dataset Type	Patients	Eyes
Total (EviRed dataset)	444	875
Test set (for all fusion tests)	53	97
Dataset 6 × 6 (for fusion of 6×6 mm${}^{2}$ OCTA: Structure + Flow)	386	753
Dataset 15 × 15 (for fusion of 15×15 mm${}^{2}$ OCTA: Structure + Flow)	372	701
Dataset 6 × 6 + 15 × 15 (for fusion of 6×6 mm${}^{2}$ + 15×15 mm${}^{2}$ OCTA)	364	676

**Table 2 diagnostics-13-02770-t002:** Distribution of eyes with different levels of severity in different datasets.

Severity	Dataset 6 × 6	Dataset 15 × 15	Dataset 6 × 6 + 15 × 15	Test Set
Absence of diabetic retinopathy	151	128	127	17
Mild NPDR	76	69	68	12
Moderate NPDR	348	334	321	39
Severe NPDR	111	107	97	18
PDR	20	20	20	3
PRP	47	43	43	8

**Table 3 diagnostics-13-02770-t003:** The results of the different data cropping methods on the test set for the input fusion of ResNet with the 15×15 mm${}^{2}$ SS-OCTA images. The best results are in bold.

Data Cropping Method	Kappa	AUC0	AUC1	AUC2	AUC3
Resize	0.2913	0.6485	0.6557	0.6836	0.7074
Center Crop	0.3270	0.7257	0.7059	0.6850	0.6903
Subvolume Crop	0.4048	0.7596	0.7429	0.7449	0.8340
*N* times Random Crop (proposed)	**0.4252**	**0.7721**	**0.7474**	**0.7519**	**0.8546**

**Table 4 diagnostics-13-02770-t004:** Backbone test results with different modalities on the test set.

Modality	Backbone	Kappa	AUC0	AUC1	AUC2	AUC3
	**ResNet**	**0.4150**	**0.8375**	**0.7659**	**0.7889**	**0.8104**
6×6 mm${}^{2}$ SS-OCTA—Structure	DenseNet	0.3597	0.8285	0.7462	0.7368	0.7040
	EfficientNet	0.4149	0.8246	0.7521	0.7438	0.7788
	**ResNet**	0.3768	0.7931	0.7653	**0.7566**	**0.7863**
6×6 mm${}^{2}$ SS-OCTA—Flow	DenseNet	0.3399	0.7972	0.7700	0.7525	0.7653
	**EfficientNet**	**0.4085**	**0.8306**	**0.7775**	0.7446	0.7150
	**ResNet**	**0.3900**	0.8118	**0.7604**	0.7462	0.8700
15×15 mm${}^{2}$ SS-OCTA—Structure	**DenseNet**	0.3589	**0.8251**	0.7527	**0.7923**	**0.8732**
	EfficientNet	0.3230	0.8046	0.7407	0.7757	0.8671
	**ResNet**	**0.4189**	**0.7927**	**0.7627**	**0.7911**	**0.8774**
15×15 mm${}^{2}$ SS-OCTA—Flow	DenseNet	0.3261	0.7770	0.7517	0.7788	0.8125
	EfficientNet	0.3259	0.7848	0.7557	0.7545	0.8397

**Table 5 diagnostics-13-02770-t005:** Results of Structure + Flow fusion for 6×6 mm${}^{2}$ SS-OCTA acquisitions on the test set. The unimodal results are baselines derived from the previous step.

Fusion Method	Backbone	Kappa	AUC0	AUC1	AUC2	AUC3
Structure (unimodal)	ResNet	0.4150	0.8375	0.7659	**0.7889**	**0.8104**
Flow (unimodal)	ResNet	0.3768	0.7931	0.7653	0.7566	0.7863
Flow (unimodal)	EfficientNet	0.4085	0.8306	0.7775	0.7446	0.7150
Input Fusion	ResNet	0.3849	0.8093	0.7656	0.7476	0.7886
Input Fusion	EfficientNet	0.3885	0.8192	0.7755	0.7496	0.7321
Feature Fusion	ResNet + ResNet	0.4329	0.8246	0.7763	0.7577	0.7900
Feature Fusion	ResNet + EfficientNet	0.3959	0.8132	0.7637	0.7023	0.7622
Decision Fusion	ResNet + ResNet	0.3814	0.8074	0.7757	0.7530	0.7868
Decision Fusion	ResNet + EfficientNet	0.4227	0.8446	0.7770	0.7500	0.7478
Hierarchical Fusion	ResNet + ResNet	**0.4752**	**0.8462**	**0.7793**	0.7607	0.8013
Hierarchical Fusion	ResNet + EfficientNet	0.4205	0.8206	0.7662	0.7186	0.7743

**Table 6 diagnostics-13-02770-t006:** Results of Structure + Flow fusion for 15×15 mm${}^{2}$ SS-OCTA images on the test set. The unimodal results are baselines derived from the previous step.

Fusion Method	Backbone	Kappa	AUC0	AUC1	AUC2	AUC3
Structure (unimodal)	ResNet	0.3900	0.8118	0.7604	0.7462	0.8700
Structure (unimodal)	DenseNet	0.3589	**0.8251**	0.7527	0.7923	0.8732
Flow (unimodal)	ResNet	0.4189	0.7927	0.7627	0.7911	0.8774
Input Fusion	ResNet	0.4252	0.7721	0.7475	0.7519	0.8546
Input Fusion	DenseNet	0.3286	0.7108	0.7072	0.7235	0.8175
Feature Fusion	ResNet + ResNet	0.3982	0.8029	0.7627	0.7876	0.8630
Feature Fusion	DenseNet + ResNet	0.3227	0.7437	0.7366	0.7546	0.8429
Decision Fusion	ResNet + ResNet	0.4124	0.7949	0.7688	0.7688	0.8728
Decision Fusion	DenseNet + ResNet	0.4376	0.8205	0.7583	0.7726	0.8754
Hierarchical Fusion	ResNet + ResNet	**0.4430**	0.8187	**0.7745**	**0.7967**	**0.8786**
Hierarchical Fusion	DenseNet + ResNet	0.4137	0.8088	0.7662	0.7794	0.8719

**Table 7 diagnostics-13-02770-t007:** Results of the 6×6 mm${}^{2}$ SS-OCTA + 15×15 mm${}^{2}$ SS-OCTA fusion on the test set. The 6×6 mm${}^{2}$ SS-OCTA and 15×15 mm${}^{2}$ SS-OCTA rows show the best performance of single acquisitions on different tasks.

Modality	Fusion Method	Kappa	AUC0	AUC1	AUC2	AUC3	Inference Time (seconds/eye)
6×6 mm${}^{2}$ SS-OCTA	Structure (unimodal)	0.4150	0.8375	0.7659	**0.7889**	**0.8104**	0.9729
	Hierarchical Fusion	**0.4752**	**0.8462**	**0.7793**	0.7607	0.8013	1.8041
15×15 mm${}^{2}$ SS-OCTA	Structure (unimodal)	0.3589	**0.8251**	0.7527	0.7923	0.8732	1.6394
	Hierarchical Fusion	**0.4430**	0.8187	**0.7745**	**0.7967**	**0.8786**	3.84655
6×6 mm${}^{2}$ SS-OCTA + 15×15 mm${}^{2}$ SS-OCTA	Feature Fusion— fine-tuning	0.4637	0.8469	0.8004	0.7989	0.8670	4.9233
	Feature Fusion— freezing layers	0.5132	0.8741	0.7853	0.7555	0.8207	4.8410
	Decision Fusion—max	0.5218	0.8801	0.8027	0.8083	0.8911	4.0686
	Decision Fusion—avg (proposed hybrid fusion)	**0.5593**	**0.8868**	**0.8276**	**0.8367**	**0.9070**	3.9679

**Table 8 diagnostics-13-02770-t008:** Results for 3D ViT with different modalities on the test set.

Modality	Backbone	Kappa	AUC0	AUC1	AUC2	AUC3
6×6 mm${}^{2}$ SS-OCTA—Structure	ViT	0.1122	0.6774	0.6490	0.4900	0.5912
6×6 mm${}^{2}$ SS-OCTA—Flow	ViT	0.0854	0.6696	0.6474	0.5487	0.5843

## Data Availability

Data are currently not publicly available due to project privacy.

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
