# Peer review of "Hybrid Fusion of High-Resolution and Ultra-Widefield OCTA Acquisitions for the Automatic Diagnosis of Diabetic Retinopathy"

_diagnostics, 2023, doi:10.3390/diagnostics13172770_

Round 1

Reviewer 1 Report

This paper proposed an innovative approach to improve the DR diagnosis by using two sets of OCTA images.  The author tested a deep learning algorithm to classify DR severity. Different fusion methods and backbones were tested and compared using clinical OCTA data. The final proposed hybrid fusion network is very interesting for the future development of deep multimodal learning in OCTA.

Some suggestions:

1.       What is the wavelength used for this OCTA?

2.       A scale bar should be added for each image in Figure 2.

3.       Regarding data cropping methods, in table 3 and Results section 3.1 data cropping, 4 different methods were compared, including resize, center crop, subvolumes crop, and N times random crop. However, in section 2.2.2 (page 6), only the first 3 were listed and explained, the proposed random crop is missed?

4.       In figure 3’s caption, the author says for the N times random cropping method, N is 10 for 6x6 mm2 OCTA, and 20 for 15x15 mm2 OCTA. Why 10 and 20? Please explain the rationale or provide some references.

Reviewer 2 Report

This work reported a deep learning based medical image processing method that can help diagnosis of diabetic retinopathy. Combining two different source data seem to naturally improve accuracy. The background is well introduced, network structures are well illustrated.

There is no fancy result from deep learning perspective, but the application on medics may play an important role. This paper is well written and organized, may be accepted after major revision as follow,

1.     Authors used classic backbone Resnet, DenseNet and EfficientNet, have authors tried other methods like VGG or more recent models like transformer and diffusion model?

2.     Authors focus on accuracy, inference speed may also influence whether or not doctors willing to use the algorithm. Would authors add inference time comparison?

Round 2

Reviewer 2 Report

The manuscript is situable for publication now.